# The Sparse Manifold Transform

**Yubei Chen**[1,2]     **Dylan M Paiton**[1,3]     **Bruno A Olshausen**[1,3,4]
[1]Redwood Center for Theoretical Neuroscience
[2]Department of Electrical Engineering and Computer Science
[3]Vision Science Graduate Group
[4]Helen Wills Neuroscience Institute & School of Optometry
University of California, Berkeley
Berkeley, CA 94720
yubeic@eecs.berkeley.edu

## Abstract

We present a signal representation framework called the *sparse manifold transform* that combines key ideas from sparse coding, manifold learning, and slow feature analysis. It turns non-linear transformations in the primary sensory signal space into linear interpolations in a representational embedding space while maintaining approximate invertibility. The sparse manifold transform is an unsupervised and generative framework that explicitly and simultaneously models the sparse discreteness and low-dimensional manifold structure found in natural scenes. When stacked, it also models hierarchical composition. We provide a theoretical description of the transform and demonstrate properties of the learned representation on both synthetic data and natural videos.

## 1   Introduction

Inspired by Pattern Theory [40], we attempt to model three important and pervasive patterns in natural signals: *sparse discreteness*, *low dimensional manifold structure* and *hierarchical composition*. Each of these concepts have been individually explored in previous studies. For example, sparse coding [43, 44] and ICA [5, 28] can learn sparse and discrete elements that make up natural signals. Manifold learning [56, 48, 38, 4] was proposed to model and visualize low-dimensional continuous transforms such as smooth 3D rotations or translations of a single discrete element. Deformable, compositional models [60, 18] allow for a hierarchical composition of components into a more abstract representation. We seek to model these three patterns jointly as they are almost always entangled in real-world signals and their disentangling poses an unsolved challenge.

In this paper, we introduce an interpretable, generative and unsupervised learning model, the *sparse manifold transform* (SMT), which has the potential to untangle all three patterns simultaneously and explicitly. The SMT consists of two stages: dimensionality expansion using sparse coding followed by contraction using manifold embedding. Our SMT implementation is to our knowledge, the first model to bridge sparse coding and manifold learning. Furthermore, an SMT layer can be stacked to produce an unsupervised hierarchical learning network.

*The primary contribution of this paper is to establish a theoretical framework for the SMT by reconciling and combining the formulations and concepts from sparse coding and manifold learning.* In the following sections we point out connections between three important unsupervised learning methods: sparse coding, local linear embedding and slow feature analysis. We then develop a single framework that utilizes insights from each method to describe our model. Although we focus here on the application to image data, the concepts are general and may be applied to other types of data such as audio signals and text. All experiments performed on natural scenes used the same dataset, described in Supplement D.

## 1.1 Sparse coding

Sparse coding attempts to approximate a data vector, $x \in \mathbb{R}^n$, as a sparse superposition of dictionary elements $\phi_i$:

$$x = \Phi \alpha + \epsilon \tag{1}$$

where $\Phi \in \mathbb{R}^{n \times m}$ is a matrix with columns $\phi_i$, $\alpha \in \mathbb{R}^m$ is a sparse vector of coefficients and $\epsilon$ is a vector containing independent Gaussian noise samples, which are assumed to be small relative to $x$. Typically $m > n$ so that the representation is *overcomplete*. For a given dictionary, $\Phi$, the sparse code, $\alpha$, of a data vector, $x$, can be computed in an online fashion by minimizing an energy function composed of a quadratic penalty on reconstruction error plus an L1 sparseness penalty on $\alpha$ (see Supplement A). The dictionary itself is adapted to the statistics of the data so as to maximize the sparsity of $\alpha$. The resulting dictionary often provides important insights about the structure of the data. For natural images, the dictionary elements become 'Gabor-like'—i.e., spatially localized, oriented and bandpass—and form a tiling over different locations, orientations and scales due to the natural transformations of objects in the world.

The sparse code of an image provides a representation that makes explicit the structure contained in the image. However the dictionary is typically *unordered*, and so the sparse code will lose the topological organization that was inherent in the image. The pioneering works of Hyvärinen and Hoyer [27], Hyvärinen et al. [29] and Osindero et al. [45] addressed this problem by specifying a fixed 2D topology over the dictionary elements that groups them according to the co-occurrence statistics of their coefficients. Other works learn the group structure from a statistical approach [37, 3, 32], but do not make explicit the underlying topological structure. Some previous topological approaches [34, 11, 10] used non-parametric methods to reveal the low-dimensional geometrical structure in local image patches, which motivated us to look for the connection between sparse coding and geometry. *From this line of inquiry, we have developed what we believe to be the first mathematical formulation for learning the general geometric embedding of dictionary elements when trained on natural scenes.*

Another observation motivating this work is that the representation computed using overcomplete sparse coding can exhibit large variability for time-varying inputs that themselves have low variability from frame to frame [49]. While some amount of variability is to be expected as image features move across different dictionary elements, the variation can appear unstructured without information about the topological relationship of the dictionary. In section 3 and section 4, we show that considering the joint spatio-temporal regularity in natural scenes can allow us to learn the dictionary's group structure and produce a representation with smooth variability from frame to frame (Figure 3).

## 1.2 Manifold Learning

In manifold learning, one assumes that the data occupy a low-dimensional, smooth manifold embedded in the high-dimensional signal space. A smooth manifold is locally equivalent to a Euclidean space and therefore each of the data points can be linearly reconstructed by using the neighboring data points. The Locally Linear Embedding (LLE) algorithm [48] first finds the neighbors of each data point in the whole dataset and then reconstructs each data point linearly from its neighbors. It then embeds the dataset into a low-dimensional Euclidean space by solving a generalized eigendecomposition problem.

The first step of LLE has the same linear formulation as sparse coding (1), with $\Phi$ being the whole dataset rather than a learned dictionary, i.e., $\Phi = X$, where $X$ is the data matrix. The coefficients, $\alpha$, correspond to the linear interpolation weights used to reconstruct a datapoint, $x$, from its $K$-nearest neighbors, resulting in a $K$-sparse code. (In other work [17], $\alpha$ is inferred by sparse approximation, which provides better separation between manifolds nearby in the same space.) Importantly, once the embedding of the dataset $X \to Y$ is computed, the embedding of a new point $x^{\text{NEW}} \to y^{\text{NEW}}$ is obtained by a simple linear projection of its sparse coefficients. That is, if $\alpha^{\text{NEW}}$ is the $K$-sparse code of $x^{\text{NEW}}$, then $y^{\text{NEW}} = Y \alpha^{\text{NEW}}$. Viewed this way, *the dictionary may be thought of as a discrete sampling of a continuous manifold, and the sparse code of a data point provides the interpolation coefficients for determining its coordinates on the manifold.* However, using the entire dataset as the dictionary is cumbersome and inefficient in practice.

Several authors [12, 53, 58] have realized that it is unnecessary to use the whole dataset as a dictionary. A random subset of the data or a set of cluster centers can be good enough to preserve the manifold structure, making learning more efficient. Going forward, we refer to these as *landmarks*. In Locally

Linear Landmarks (LLL) [58], the authors compute two linear interpolations for each data point $x$:

$$x = \Phi_{\text{LM}}\,\alpha + n \qquad (2)$$
$$x = \Phi_{\text{DATA}}\,\gamma + n' \qquad (3)$$

where $\Phi_{\text{LM}}$ is a dictionary of landmarks and $\Phi_{\text{DATA}}$ is a dictionary composed of the whole dataset. As in LLE, $\alpha$ and $\gamma$ are coefficient vectors inferred using KNN solvers (where the $\gamma$ coefficient corresponding to $x$ is forced to be 0). We can substitute the solutions to equation (2) into $\Phi_{\text{DATA}}$, giving $\Phi_{\text{DATA}} \approx \Phi_{\text{LM}}A$, where the $j^{\text{th}}$ column of the matrix $A$ is a unique vector $\alpha_j$. This leads to an interpolation relationship:

$$\Phi_{\text{LM}}\alpha \approx \Phi_{\text{LM}}\,A\,\gamma \qquad (4)$$

The authors sought to embed the landmarks into a low dimensional Euclidean space using an embedding matrix, $P_{\text{LM}}$, such that the interpolation relationship in equation (4) still holds:

$$P_{\text{LM}}\alpha \approx P_{\text{LM}}\,A\,\gamma \qquad (5)$$

Where we use the same $\alpha$ and $\gamma$ vectors that allowed for equality in equations (2) and (3). $P_{\text{LM}}$ is an embedding matrix for $\Phi_{\text{LM}}$ such that each of the columns of $P$ represents an embedding of a landmark. $P_{\text{LM}}$ can be derived by solving a generalized eigendecomposition problem [58].

The similarity between equation (1) and equation (2) provides an intuition to bring sparse coding and manifold learning closer together. However, LLL still has a difficulty in that it requires a nearest neighbor search. We posit that temporal information provides a more natural and efficient solution.

### 1.3   Slow Feature Analysis (SFA)

The general idea of imposing a 'slowness prior' was initially proposed by [20] and [59] to extract invariant or slowly varying features from temporal sequences rather than using static orderless data points. While it is still common practice in both sparse coding and manifold learning to collect data in an orderless fashion, other work has used time-series data to learn spatiotemporal representations [57, 41, 30] or to disentangle form and motion [6, 9, 13]. Specifically, the combination of topography and temporal coherence in [30] provides a strong motivation for this work.

Here, we utilize *temporal adjacency* to determine the nearest neighbors in the embedding space (eq. 3) by specifically minimizing the second-order temporal derivative, implying that video sequences form *linear trajectories* in the manifold embedding space. A similar approach was recently used by [23] to linearize transformations in natural video. This is a variation of 'slowness' that makes the connection to manifold learning more explicit. It also connects to the ideas of manifold flattening [14] or straightening [24] which are hypothesized to underly perceptual representations in the brain.

## 2   Functional Embedding: A Sensing Perspective

The SMT framework differs from the classical manifold learning approach in that it relies on the concept of *functional embedding* as opposed to embedding individual data points. We explain this concept here before turning to the sparse manifold transform in section 3.

In classical manifold learning [26], for a $m$-dimensional compact manifold, it is typical to solve a generalized eigenvalue decomposition problem and preserve the $2^{\text{nd}}$ to the $(d + 1)^{\text{th}}$ trailing eigenvectors as the embedding matrix $P_{\text{C}} \in \mathbb{R}^{d\times N}$, where $d$ is as small as possible (parsimonious) such that the embedding preserves the topology of the manifold (usually, $m \le d \le 2m$ due to the strong Whitney embedding theorem[35]) and $N$ is the number of data points or landmarks to embed. It is conventional to view the columns of an embedding matrix, $P_{\text{C}}$, as an embedding to an Euclidean space, which is (at least approximately) topologically-equivalent to the data manifold. Each of the rows of $P_{\text{C}}$ is treated as a coordinate of the underlying manifold. One may think of a point on the manifold as a single, constant-amplitude delta function with the manifold as its domain. Classical manifold embedding turns a non-linear transformation (i.e., a moving delta function on the manifold) in the original signal space into a simple linear interpolation in the embedding space. This approach is effective for visualizing data in a low-dimensional space and compactly representing the underlying geometry, but less effective when the underlying function is not a single delta function.

In this work we seek to move beyond the single delta-function assumption, because natural images are not well described as a single point on a continuous manifold of fixed dimensionality. For any

reasonably sized image region (e.g., a $16 \times 16$ pixel image patch), there could be multiple edges moving in different directions, or the edge of one occluding surface may move over another, or the overall appearance may change as features enter or exit the region. Such changes will cause the manifold dimensionality to vary substantially, so that the signal structure is no longer well-characterized as a manifold.

We propose instead to think of any given image patch as consisting of $h$ discrete components simultaneously moving over the same underlying manifold - i.e., as $h$ delta functions, or *an h-sparse function* on the smooth manifold. This idea is illustrated in figure 1. First, let us organize the Gabor-like dictionary learned from natural scenes on a 4-dimensional manifold according to the position $(x, y)$, orientation $(\theta)$ and scale $(\sigma)$ of each dictionary element $\phi_i$. Any given Gabor function corresponds to a point with coordinates $(x, y, \theta, \sigma)$ on this manifold, and so the learned dictionary as a whole may be conceptualized as a discrete tiling of the manifold. Then, the $k$-sparse code of an image, $\alpha$, can be viewed as a set of $k$ delta functions on this manifold (illustrated as black arrows in figure 1C). Hyvärinen has pointed out that when the dictionary is topologically organized in a similar manner, the active coefficients $\alpha_i$ tend to form clusters, or "bubbles," over this domain [30]. Each of these clusters may be thought of as linearly approximating a "virtual Gabor" at the center of the cluster (illustrated as red arrows in figure 1C), effectively performing a flexible "steering" of the dictionary to describe discrete components in the image, similar to steerable filters [21, 55, 54, 47]. Assuming there are $h$ such clusters, then the $k$-sparse code of the image can be thought of as a discrete approximation of an underlying $h$-sparse function defined on the continuous manifold domain, where $h$ is generally greater than 1 but less than $k$.

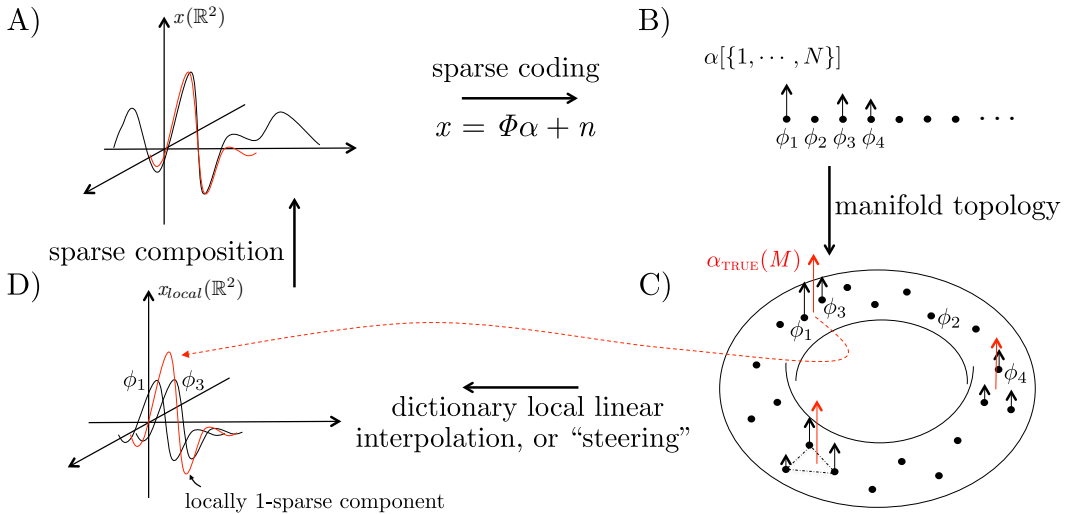

Figure 1: Dictionary elements learned from natural signals with sparse coding may be conceptualized as landmarks on a smooth manifold. A) A function defined on $\mathbb{R}^2$ (e.g. a gray-scale natural image) and one local component from its reconstruction are represented by the black and red curves, respectively. B) The signal is encoded using sparse inference with a learned dictionary, $\Phi$, resulting in a $k$-sparse vector (also a function) $\alpha$, which is defined on an orderless discrete set $\{1, \cdots, N\}$. C) $\alpha$ can be viewed as a discrete $k$-sparse approximation to the true $h$-sparse function, $\alpha_{\text{TRUE}}(M)$, defined on the smooth manifold ($k = 8$ and $h = 3$ in this example). Each dictionary element in $\Phi$ corresponds to a landmark (black dot) on the smooth manifold, $M$. Red arrows indicate the underlying $h$-sparse function, while black arrows indicate the $k$ non-zero coefficients of $\Phi$ used to interpolate the red arrows. D) Since $\Phi$ only contains a finite number of landmarks, we must interpolate (i.e. "steer") among a few dictionary elements to reconstruct each of the true image components.

An $h$-sparse function would not be recoverable from the $d$-dimensional projection employed in the classical approach because the embedding is premised on there being only a single delta function on the manifold. Hence the inverse will not be uniquely defined. Here we utilize a more general *functional* embedding concept that allows for better recovery capacity. A functional embedding of the landmarks is to take the first $f$ trailing eigenvectors from the generalized eigendecomposition solution

as the embedding matrix $P \in \mathbb{R}^{f \times N}$, where $f$ is larger than $d$ such that the $h$-sparse function can be recovered from the linear projection. Empirically[1] we use $f = O(h \log(N))$.

To illustrate the distinction between the classical view of a data manifold and the additional properties gained by a functional embedding, let us consider a simple example of a function over the 2D unit disc. Assume we are given 300 landmarks on this disc as a dictionary $\Phi_{\mathrm{LM}} \in \mathbb{R}^{2 \times 300}$. We then generate many short sequences of a point $x$ moving along a straight line on the unit disc, with random starting locations and velocities. At each time, $t$, we use a nearest neighbor (KNN) solver to find a local linear interpolation of the point's location from the landmarks, that is $x_t = \Phi_{\mathrm{LM}} \alpha_t$, with $\alpha_t \in \mathbb{R}^{300}$ and $\alpha_t \succeq 0$ (the choice of sparse solver does not impact the demonstration). Now we seek to find an embedding matrix, $P$, which projects the $\alpha_t$ into an $f$-dimensional space via $\beta_t = P \alpha_t$ such that the trajectories in $\beta_t$ are as straight as possible, thus reflecting their true underlying geometry. This is achieved by performing an optimization that minimizes the second temporal derivative of $\beta_t$, as specified in equation (8) below.

Figure 2A shows the rows of $P$ resulting from this optimization using $f = 21$. Interestingly, they resemble Zernike polynomials on the unit-disc. We can think of these as functionals that "sense" sparse functions on the underlying manifold. Each row $p'_i \in \mathbb{R}^{300}$ (here the prime sign denotes a row of the matrix $P$) projects a discrete $k$-sparse approximation $\alpha$ of the underlying $h$-sparse function to a real number, $\beta_i$. We define the full set of these linear projections $\beta = P \alpha$ as a "manifold sensing" of $\alpha$.

When there is only a single delta-function on the manifold, the second and third rows of $P$, which form simple linear ramp functions in two orthogonal directions, are sufficient to fully represent its position. These two rows would constitute $P_{\mathrm{C}} \in \mathbb{R}^{2 \times 300}$ as an embedding solution in the classical manifold learning approach, since a unit disk is diffeomorphic to $\mathbb{R}^2$ and can be embedded in a 2 dimensional space. The resulting embedding $\beta_2, \beta_3$ closely resembles the 2-D unit disk manifold and allows for recovery of a one-sparse function, as shown in Figure 2B.

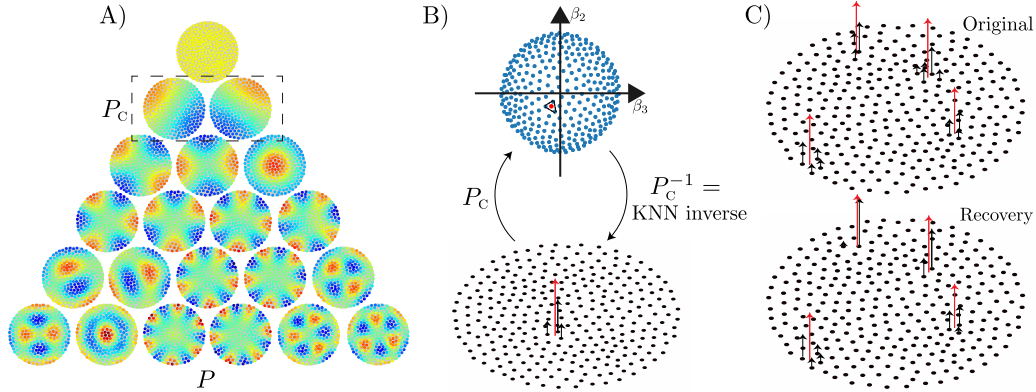

Figure 2: Demonstration of functional embedding on the unit disc. A) The rows of $P$, visualized here on the ground-truth unit disc. Each disc shows the weights in a row of $P$ by coloring the landmarks according to the corresponding value in that row of $P$. The color scale for each row is individually normalized to emphasize its structure. The pyramidal arrangement of rows is chosen to highlight their strong resemblance to the Zernike polynomials. B) (Top) The classic manifold embedding perspective allows for low-dimensional data visualization using $P_{\mathrm{C}}$, which in this case is given by the second and third rows of $P$ (shown in dashed box in panel A). Each blue dot shows the 2D projection of a landmark using $P_{\mathrm{C}}$. Boundary effects cause the landmarks to cluster toward the perimeter. (Bottom) A 1-sparse function is recoverable when projected to the embedding space by $P_{\mathrm{C}}$. C) (Top) A 4-sparse function (red arrows) and its discrete $k$-sparse approximation, $\alpha$ (black arrows) on the unit disc. (Bottom) The recovery, $\alpha_{\mathrm{REC}}$, (black arrows) is computed by solving the optimization problem in equation (6). The estimate of the underlying function (red arrows) was computed by taking a normalized local mean of the recovered $k$-sparse approximations for a visualization purpose.

Recovering more than a one-sparse function requires using additional rows of $P$ with higher spatial-frequencies on the manifold, which together provide higher sensing capacity. Figure 2C demonstrates

recovery of an underlying 4-sparse function on the manifold using all 21 functionals, from $p_1'$ to $p_{21}'$. From this representation, we can recover an estimate of $\alpha$ with positive-only sparse inference:

$$\alpha_{\text{REC}} = g(\beta) \equiv \underset{\alpha}{\operatorname{argmin}} \|\beta - P\alpha\|_F^2 + \lambda z^T \alpha, \ \ \text{s.t. } \alpha \succeq 0, \tag{6}$$

where $z = [\|p_1\|_2, \cdots, \|p_N\|_2]^T$ and $p_j \in \mathbb{R}^{21}$ is the $j^{\text{th}}$ column of $P$. Note that although $\alpha_{\text{REC}}$ is not an exact recovery of $\alpha$, the 4-sparse structure is still well preserved, up to a local shift in the locations of the delta functions. We conjecture this will lead to a recovery that is perceptually similar for an image signal.

The functional embedding concept can be generalized beyond functionals defined on a single manifold and will still apply when the underlying geometrical domain is a union of several different manifolds. A thorough analysis of the capacity of this sensing method is beyond the scope of this paper, although we recognize it as an interesting research topic for model-based compressive sensing.

## 3 The Sparse Manifold Transform

The Sparse Manifold Transform (SMT) consists of a non-linear sparse coding expansion followed by a linear manifold sensing compression (dimension reduction). The manifold sensing step acts to linearly pool the sparse codes, $\alpha$, with a matrix, $P$, that is learned using the functional embedding concept (sec. 2) in order to straighten trajectories arising from video (or other dynamical) data.

The SMT framework makes three basic assumptions:

1. The dictionary $\Phi$ learned by sparse coding has an organization that is a discrete sampling of a low-dimensional, smooth manifold, $M$ (Fig. 1).

2. The resulting sparse code $\alpha$ is a discrete $k$-sparse approximation of an underlying $h$-sparse function defined on $M$. There exists a functional manifold embedding, $\tau : \Phi \hookrightarrow P$, that maps each of the dictionary elements to a new vector, $p_j = \tau(\phi_j)$, where $p_j$ is the $j^{\text{th}}$ column of $P$ s.t. both the topology of $M$ and $h$-sparse function's structure are preserved.

3. A continuous temporal transformation in the input (e.g., from natural movies) lead to a linear flow on $M$ and also in the geometrical embedding space.

In an image, the elements of the underlying $h$-sparse function correspond to discrete components such as edges, corners, blobs or other features that are undergoing some simple set of transformations. Since there are only a finite number of learned dictionary elements tiling the underlying manifold, they must cooperate (or 'steer') to represent each of these components as they appear along a continuum.

The desired property of linear flow in the geometric embedding space may be stated mathematically as

$$P\alpha_t \approx \tfrac{1}{2} P\alpha_{t-1} + \tfrac{1}{2} P\alpha_{t+1}. \tag{7}$$

where $\alpha_t$ denotes the sparse coefficient vector at time $t$. Here we exploit the temporal continuity inherent in the data to solve the otherwise cumbersome nearest-neighbor search required of LLE or LLL. The embedding matrix $P$ satisfying (7) may be derived by minimizing an objective function that encourages the second-order temporal derivative of $P\alpha$ to be zero:

$$\min_P \|PAD\|_F^2, \ \text{s.t. } PVP^T = I \tag{8}$$

where $A$ is the coefficient matrix whose columns are the coefficient vectors, $\alpha_t$, in temporal order, and $D$ is the second-order differential operator matrix, with $D_{t-1,t} = -0.5, D_{t,t} = 1, D_{t+1,t} = -0.5$ and $D_{\tau,t} = 0$ otherwise. $V$ is a positive-definite matrix for normalization, $I$ is the identity matrix and $\| \bullet \|_F$ indicates the matrix Frobenius norm. We choose $V$ to be the covariance matrix of $\alpha$ and thus the optimization constraint makes the rows of $P$ orthogonal in whitened sparse coefficient vector space. Note that this formulation is qualitatively similar to applying SFA to sparse coefficients, but using the second-order derivative instead of the first-order derivative.

The solution to this generalized eigen-decomposition problem is given [58] by $P = V^{-\frac{1}{2}} U$, where $U$ is a matrix of $f$ trailing eigenvectors (i.e. eigenvectors with the smallest eigenvalues) of the matrix $V^{-\frac{1}{2}} ADD^T A^T V^{-\frac{1}{2}}$. Some drawbacks of this analytic solution are that: 1) there is an unnecessary ordering among different dimensions, 2) the learned functional embedding tends to be global, which

has support as large as the whole manifold and 3) the solution is not online and does not allow other constraints to be posed. In order to solve these issues, we modify the formulation slightly with a sparse regularization term on $P$ and develop an online SGD (Stochastic Gradient Descent) solution, which is detailed in the Supplement C.

To summarize, the SMT is performed on an input signal $x$ by first computing a higher-dimensional representation $\alpha$ via sparse inference with a learned dictionary, $\Phi$, and second computing a contracted code by sensing a manifold representation, $\beta = P\alpha$ with a learned pooling matrix, $P$.

## 4    Results

**Straightening of video sequences.**  We applied the SMT optimization procedure on sequences of whitened $20 \times 20$ pixel image patches extracted from natural videos. We first learned a $10\times$ overcomplete spatial dictionary $\Phi \in \mathbb{R}^{400\times 4000}$ and coded each frame $x_t$ as a 4000-dimensional sparse coefficient vector $\alpha_t$. We then derived an embedding matrix $P \in \mathbb{R}^{200\times 4000}$ by solving equation 8. Figure 3 shows that while the sparse code $\alpha_t$ exhibits high variability from frame to frame, the embedded representation $\beta_t = P\alpha_t$ changes in a more linear or smooth manner. It should be emphasized that finding such a smooth linear projection (embedding) is highly non-trivial, and is possible if and only if the sparse codes change in a locally linear manner in response to smooth transformations in the image. If the sparse code were to change in an erratic or random manner under these transformations, any linear projection would be non-smooth in time. Furthermore, we show that this embedding does not constitute a trivial temporal smoothing, as we can recover a good approximation of the image sequence via $\hat{x}_t = \Phi\, g(\beta_t)$, where $g(\beta)$ is the inverse embedding function (6). We can also use the functional embedding to regularize sparse inference, as detailed in Supplement B, which further increases the smoothness of both $\alpha$ and $\beta$.

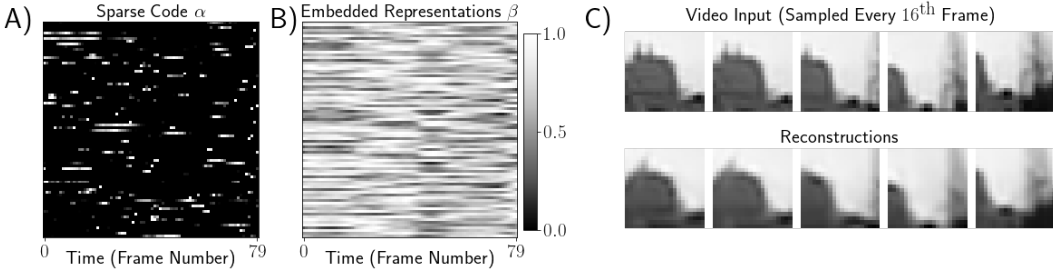

Figure 3: SMT encoding of a 80 frame image sequence. A) Rescaled activations for 80 randomly selected $\alpha$ units. Each row depicts the temporal sequence of a different unit. B) The activity of 80 randomly selected $\beta$ units. C) Frame samples from the 90fps video input (top) and reconstructions computed from the $\alpha_{\text{REC}}$ recovered from the sequence of $\beta$ values (bottom).

**Affinity Groups and Dictionary Topology.**  Once a functional embedding is learned for the dictionary elements, we can compute the cosine similarity between their embedding vectors, $\cos(p_j, p_k) = \frac{p_j^T p_k}{\|p_j\|_2 \|p_k\|_2}$, to find the neighbors, or affinity group, of each dictionary element in the embedding space. In Figure 4A we show the affinity groups for a set of randomly sampled elements from the overcomplete dictionary learned from natural videos. As one can see, the topology of the embedding learned from the SMT reflects the structural similarity of the dictionary elements according to the properties of position, orientation, and scale. Figure 4B shows that the nearest neighbors of each dictionary element in the embedding space are more 'semantically similar' than the nearest neighbors of the element in the pixel space. To measure the similarity, we chose the top 500 most well-fit dictionary elements and computed their lengths and orientations. For each of these elements, we find the top 9 nearest neighbors in both the embedding space and in pixel space and then compute the average difference in length ($\Delta$ Length) and orientation ($\Delta$ Angle). The results confirm that the embedding space is succeeding in grouping dictionary elements according to their structural similarity, presumably due to the continuous geometric transformations occurring in image sequences.

Computing the cosine similarity can be thought of as a hypersphere normalization on the embedding matrix $P$. In other words, if the embedding is normalized to be approximately on a hypersphere,

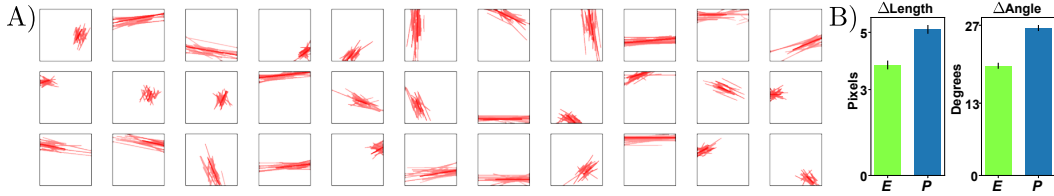

Figure 4: A) Affinity groups learned using the SMT reveal the topological ordering of a sparse coding dictionary. Each box depicts as a needle plot the affinity group of a randomly selected dictionary element and its top 40 affinity neighbors. The length, position, and orientation of each needle reflect those properties of the dictionary element in the affinity group (see Supplement E for details). The color shade indicates the normalized strength of the cosine similarity between the dictionary elements. B) The properties of length and orientation (angle) are more similar among nearest neighbors in the embedding space ($E$) as compared to the pixel space ($P$).

the cosine distance is almost equivalent to the Gramian matrix, $P^T P$. Taking this perspective, the learned geometric embedding and affinity groups can explain the dictionary grouping results shown in previous work [25]. In that work, the layer 1 outputs are pooled by an affinity matrix given by $P = E^T E$, where $E$ is the eigenvector matrix computed from the correlations among layer 1 outputs. This PCA-based method can be considered an embedding that uses only spatial correlation information, while the SMT model uses both spatial correlation and temporal interpolation information.

**Hierarchical Composition.** A SMT layer is composed of two sublayers: a sparse coding sublayer that models sparse discreteness, and a manifold embedding sublayer that models simple geometrical transforms. It is possible to stack multiple SMT layers to form a hierarchical architecture, which addresses the third pattern from Mumford's theory: hierarchical composition. It also provides a way to progressively flatten image manifolds, as proposed by DiCarlo & Cox [14]. Here we demonstrate this process with a two-layer SMT model (Figure 5A) and we visualize the learned representations. The network is trained in a layer-by-layer fashion on a natural video dataset as above.

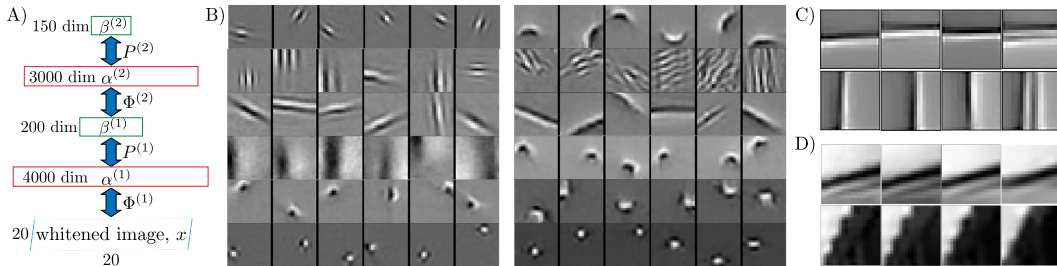

Figure 5: SMT layers can be stacked to learn a hierarchical representation. A) The network architecture. Each layer contains a sparse coding sublayer (red) and a manifold sensing sublayer (green). B) Example dictionary element groups for $\Phi^{(1)}$ (left) and $\Phi^{(2)}$ (right). C) Each row shows an example of interpolation by combining layer 3 dictionary elements. From left to right, the first two columns are visualizations of two different layer-3 dictionary elements, each obtained by setting a single element of $\alpha^{(3)}$ to one and the rest to zero. The third column is an image generated by setting both elements of $\alpha^{(3)}$ to 0.5 simultaneously. The fourth column is a linear interpolation in image space between the first two images, for comparison. D) Information is approximately preserved at higher layers. From left to right: The input image and the reconstructions from $\alpha^{(1)}$, $\alpha^{(2)}$ and $\alpha^{(3)}$, respectively. The rows in C) and D) are unique examples. See section 2 for visualization details.

We can produce reconstructions and dictionary visualizations from any layer by repeatedly using the inverse operator, $g(\beta)$. Formally, we define $\alpha_{\text{REC}}^{(l)} = g^{(l)}(\beta^{(l)})$, where $l$ is the layer number. For example, the inverse transform from $\alpha^{(2)}$ to the image space will be $x_{\text{REC}} = C\Phi^{(1)}g^{(1)}(\Phi^{(2)}\alpha^{(2)})$, where $C$ is an unwhitening matrix. We can use this inverse transform to visualize any single dictionary element by setting $\alpha^{(l)}$ to a 1-hot vector. Using this method of visualization, Figure 5B shows a comparison of some of the dictionary elements learned at layers 1 and 2. We can see that lower layer

elements combine together to form more global and abstract dictionary elements in higher layers, e.g. layer-2 units tend to be more curved, many of them are corners, textures or larger blobs.

Another important property that emerges at higher levels of the network is that dictionary elements are steerable over a larger range, since they are learned from progressively more linearized representations. To demonstrate this, we trained a three-layer network and performed linear interpolation between two third-layer dictionary elements, resulting in a non-linear interpolation in the image space that shifts features far beyond what simple linear interpolation in the image space would accomplish (Figure 5C). A thorough visualization of the dictionary elements and groups is provided in the Supplement F.

## 5  Discussion

A key new perspective introduced in this work is to view both the signals (such as images) and their sparse representations as functions defined on a manifold domain. A gray-scale image is a function defined on a 2D plane, tiled by pixels. Here we propose that the dictionary elements should be viewed as the new 'pixels' and their coefficients are the corresponding new 'pixel values'. The pooling functions can be viewed as low pass filters defined on this new manifold domain. This perspective is strongly connected to the recent development in both signal processing on irregular domains [52] and geometric deep learning [7].

Previous approaches have learned the group structure of dictionary elements mainly from a statistical perspective [27, 29, 45, 32, 37, 39]. Additional unsupervised learning models [51, 46, 33, 62] combine sparse discreteness with hierarchical structure, but do not explicitly model the low-dimensional manifold structure of inputs. Our contribution here is to approach the problem from a geometric perspective to learn a topological embedding of the dictionary elements.

The functional embedding framework provides a new perspective on the pooling functions commonly used in convnets. In particular, it provides a principled framework for learning the pooling operators at each stage of representation based on the underlying geometry of the data, rather than being imposed in a 2D topology *a priori* as was done previously to learn linearized representations from video [23]. This could facilitate the learning of higher-order invariances, as well as equivariant representations [50], at higher stages. In addition, since the pooling is approximately invertible due to the underlying sparsity, it is possible to have bidirectional flow of information between stages of representation to allow for hierarchical inference [36]. The invertibility of SMT is due to the underlying sparsity of the signal, and is related to prior works on the invertibility of deep networks[22, 8, 61, 16]. Understanding this relationship may bring further insights to these models.

**Acknowledgments**

We thank Joan Bruna, Fritz Sommer, Ryan Zarcone, Alex Anderson, Brian Cheung and Charles Frye for many fruitful discussions; Karl Zipser for sharing computing resources; Eero Simoncelli and Chris Rozell for pointing us to some valuable references. This work is supported by NSF-IIS-1718991, NSF-DGE-1106400, and NIH/NEI T32 EY007043.

## Footnotes

[1]This choice is inspired by the result from compressive sensing[15], though here $h$ is different from $k$.

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
