[Supplementary Material]

# Supplementary Material

## A  Sparse Coding

In traditional sparse coding [43] approach, the coefficient vector $\alpha$ is computed for each input image by performing the optimization:

$$\min_{\alpha} \tfrac{1}{2}\|x - \Phi\alpha\|_2^2 + \lambda\|\alpha\|_1 \tag{9}$$

where $\lambda$ is a sparsity trade-off penalty.

In this paper we learn a 10-20 times overcomplete dictionary [42] and impose the additional restriction of positive-only coefficients ($\alpha \succeq 0$). Empirically, we find that at such an overcompleteness the interpolation behavior of the dictionary is close to locally linear. Therefore, steering the elements can be accomplished by local neighborhood interpolation. High overcompleteness with a positive-only constraint makes the relative geometry of the dictionary elements more explicit. The positive-only constraint gives us our modified optimization:

$$\min_{\alpha} \tfrac{1}{2}\|x - \Phi\alpha\|_2^2 + \lambda\|\alpha\|_1, \text{ s.t.} \alpha \succeq 0, \tag{10}$$

## B  Sparse Coefficient Inference with Manifold Embedding Regularization

Functional manifold embedding can be used to regularize sparse inference for video sequences. We assume that in the embedding space the representation has a locally linear trajectory. Then we can change the formulation from equation 10 to the following:

$$
\begin{aligned}
\min_{\alpha_t} \quad & \|x_t - \Phi\alpha_t\|_2^2 + \lambda\|\alpha_t\|_1 + \gamma_0\|P(\alpha_t - \widetilde{\alpha}_t)\|_2^2 \\
\text{s.t.} \quad & \alpha_t \succeq 0, \; \widetilde{\alpha}_t = \alpha_{t-1} + (\alpha_{t-1} - \alpha_{t-2}),
\end{aligned}
\tag{11}
$$

where $\widetilde{\alpha}_t$ is the casual linear prediction from the previous two steps $\alpha_{t-1}$, $\alpha_{t-2}$. Our preliminary experiments indicate that this can significantly increase the linearity of the embedding $\beta = P\alpha$.

## C  Online SGD Solution

Some minor drawbacks of the analytic generalized eigendecomposition solution are that: 1) there is an unnecessary ordering among different embedding dimensions, 2) the learned functional embedding tends to be global, which has support as large as the whole manifold and 3) the solution is not online and does not allow other constraints to be posed. In order to solve these issues, we modify the formulation of equation 8 slightly with a sparse regularization term on $P$ and develop an online SGD (Stochastic Gradient Descent) solution:

$$\min_{P} \gamma_0\|PAD\|_F^2 + \gamma_1\|PW\|_1, \text{ s.t. } PVP^T = I, \tag{12}$$

where $\gamma_0$ and $\gamma_1$ are both positive parameters, $W = \text{diag}(\langle\alpha\rangle)$ and $\|\bullet\|_1$ is the $L_{1,1}$ norm, $V$ is the covariance matrix of $\alpha$. The sparse regularization term encourages the functionals to be localized on the manifold.

The iterative update rule for solving equation (12) to learn the manifold embedding consists of:

1. One step along the whitened gradient computed on a mini-batch: $P \mathrel{+}= -2\gamma_0\eta_P PADD^T A^T V^{-1}$, where $V^{-1}$ serves to whiten the gradient. $\eta_P$ is the learning rate of $P$.

2. Weight regularization: $P = \text{Shrinkage}_{\langle\alpha\rangle, \gamma1}(P)$, which shrinks each entry in the $j^{\text{th}}$ column of $P$ by $\gamma_1\langle\alpha_j\rangle$.

3. Parallel orthogonalization: $(PVP^T)^{-\frac{1}{2}}P$, which is a slight variation to an orthogonalization method introduced by [31].

## D Dataset and Preprocessing

The natural videos were captured with a head-mounted GoPro 5 session camera at 90fps. The video dataset we used contains about 53 minutes (about 300,000 frames) of 1080p video while the cameraperson is walking on a college campus. For each video frame, we select the center 1024x1024 section and down-sample it to 128x128 by using a steerable pyramid [54] to decrease the range of motion in order to avoid temporal aliasing. For each 128x128 video frame, we then apply an approximate whitening in the Fourier domain: For each frame, 1) take a Fourier transform, 2) modulate the amplitude by a whitening mask function $w(u, v)$, 3) take an inverse Fourier transform. Since the Fourier amplitude map of natural images in general follows a $1/f$ law[19], the whitening mask function is chosen to be the product: $w(u, v) = w_1(u, v)w_2(u, v)$, where $w_1(u, v) = r$ is a linear ramp function of the frequency radius $r(u, v) = (u^2 + v^2)^{\frac{1}{2}}$ and $w_2$ is a low-pass windowing function in the frequency domain $w_2(u, v) = e^{-(\frac{r}{r_0})^4}$, with $r_0 = 48$. The resulting amplitude modulation function is shown in Supplementary Figure 1. For a more thorough discussion on this method, please see [1, 2, 44]. We found that the exact parameters for whitening were not critical. We also implemented a ZCA version and found that the results were qualitatively similar.

Supplementary Figure 1: In this figure we show the whitening mask function $w(u, v)$. Since $w$ is radial symmetric, we show its value with respect to the frequency radius $r$ as a 1-D curve. As we are using 128x128 images, the frequency radius we show is from 0 to 64, which is the Nyquist frequency.

## E Needle Plot Details

Figure 4 utilized a visualization technique that depicts each dictionary element as a needle whose position, orientation and length indicate those properties of the element. The parameters of the needles were computed from a combination of the analytic signal envelope and the Fourier transform of each dictionary element. The envelope was fitted with a 2D Gaussian, and its mean was used to determine the center location of the dictionary element. The primary eigenvector of the Gaussian covariance matrix was used to determine the spatial extent of the primary axis of variation, which we indicated by the bar length. The orientation was computed from the peak in the Fourier amplitude map of the dictionary element. Supplementary Figure 2 gives an illustration of the fitting process. A subset of elements were not well summarized by this methodology because they do not fit a standard Gaussian profile, which can be seen in the last row of Supplementary Figure 2. However, the majority appear to be well characterized by this method. Each box in the main-text figure 4 indicates a single affinity group. The color indicates the normalized similarity between the dictionary elements in the embedding space.

## F Dictionary Visualization

Here we provide a more complete visualization of the dictionary elements in the two-layer SMT model. In Supplementary Figure 3, we show 19 representative dictionary elements and their top

Supplementary Figure 2: Needle fitting procedure

5 affinity group neighbors in both layer 1 (Supplementary Figure 3A) and layer 2 (Supplementary Figure 3B). For each row, we choose a representative dictionary element (on the left side) and compute its cosine similarity to the rest of the dictionary elements in the embedding space. The closest 5 elements in the embedding space are shown on its right. So each row can be treated as a small affinity group. Figures 4 and 5 show a large random sample of the dictionary elements in layers 1 and 2, respectively.

Supplementary Figure 3: Representative dictionary elements and their top 5 affinity group neighbors in the embedding space. A) Dictionary elements from layer 1. B) Dictionary elements from layer 2. We encourage the reader to compare the affinity neighbors to the results found in [61].

Supplementary Figure 4: A random sample of 1200 dictionary elements from layer 1 (out of 4000 units). Details are better discerned by zoomed in on a monitor.

Supplementary Figure 5: A random sample of 1200 dictionary elements from layer 2 (out of 3000 units). Details are better discerned by zoomed in on a monitor.