[Reviews · NeurIPS 2018]

Reviewer 1



This paper presents an unsupervised learning method called sparse manifold transform for learning low-dimensional structures from data. The proposed approach is based on combining sparse coding and manifold embedding. The properties of the learned representations are demonstrated on synthetic and real data. Overall, this paper is a long-winded presentation of a new manifold learning approach, which (at least in my view) is explained in heuristic and vague terms and seems very hard to perceive. - First of all, it is not clear what modeling of data is being considered. Conventional manifold learning makes the assumption that the high dimensional data points lie close to a low dimensional manifold, and different manifold learning methods find low-dimensional embeddings that preserve different local/global properties of data. This paper seems to be considering something more than that. It introduces a concept called h-sparse function, which has no precise definition and presumably means that each data point in the original data space correspond to multiple points on the low-dimensional manifold. If this is the case, then how are sparse coding coefficients interpreted? Should one assume that the vector of sparse coefficients in this case is the sum of sparse coefficient vectors for each of the point on the low-dimensional manifold? This seems to be illustrated in Figure 1, but it does not seem that the figure provides enough information for what is being computed in there. Another confusion is that the proposed SMT seems to be utilizing temporal smoothness of data, with movies being an example. Does this mean that SMT is designed for manifold learning of temporal data only? This should be made clear at the beginning of the paper rather than just not bringing this up until page 6 of the paper. - There are also a lot of technical details/terms which are explained only vaguely. One example is the so-called "h-sparse function" mentioned above which does not have a clear description of the concept (unless it is a well-known concept). Another example is line 141 which mentioned section 1.2 but it is unclear to me how section 1.2 is related to the context here. Also, line 150 mentioned "2D manifold", I don't see which is the manifold. There is also no formal description of their algorithm. It is said that the first step is sparse coding, but how is the dictionary learned for sparse coding? - It appears that the method is closely related to LLE and LLL, in the sense that all of them are based on data representation. In fact, the paper started with a review of LLL in Section 1.2. But later on in the description of the proposed method there does not seem to have any comment on the connections with LLL. I feel that a comment on such connection will be helpful to clarify the technical contributions of the paper.

Reviewer 2



The paper "The Sparse Manifold Transform" discusses a combination of sparse coding, SFA-like learning and manifold learning to represent sensory data (with a focus on images). The approach is motivated, derived and finally tested on video. Evaluation The paper addresses important research questions and a challenging task. The big picture question is how complex sensory stimuli can be represented using low complexity representation for the important causes of the data. The general idea is that data such as videos are ultimately caused by regular movements of usually just a few objects. If such regularities can be captured, reasoning about the data is becoming easy which makes an approach as developed here very worthy to follow. Sparse coding is traditionally seen as a first processing step capturing low-level image components such as edges, so I also think that its use here makes sense. Sparse coding is then coupled to a manifold representation to realize a low level representation of objects. This is usually difficult by itself and the generalization using sparse coding also makes sense, I believe. Finally, using temporal information also makes sense. In general, this approach is a combination of different ideas, and I like that the used concepts are not just combined in successive order but that they are coupled through changed objectives and learned simultaneously on data. To realize the whole combined approach such that it can be applied to real data and produces results is an achievement. Having said so, it is difficult to come up with a quantitative evaluation, and this difficulty is connected to how this approach shall be evaluated in comparison with other approaches. As there are different types of learning concepts involved, and as for each learning concept different alternatives could be used, a number of design decisions had to be made here. It is simply very difficult to say which decision has what kind of impact on the actual learning and results. To start with the first concept. On the one hand, a standard sparse coding approach is used, i.e., the objective in supplement eq 1 corresponds to a Laplace prior (L_1 sparsity). On the other hand, the paper emphasises the encoding of objects as discrete elements a several points. Wouldn't a sparse coding objective be more suitable that does reflect such a desired discreteness (e.g. Henniges et al., LVA 2010, Titsias, NIPS 2011, Sheikh, Lucke, NIPS 2016 or any deterministic L_0 sparse coding approach)? The main merit here is the combination of a sparse coding objective with manifold learning (eq. 2 of supplement), but there isn't anything specific to L_1 sparse coding that is conceptually important here. On the other hand, other sprase objectives may aimpact the spherical projection operator used. Shouldn't there be a hard-sparse selection here anyway before \Phi is applied? Close to zero basis functions would be amplified otherwise (but I may miss something here). Regarding manifold embeddings, the linearity assumption of points moving on the manifold makes sense for smooth changes, but especially occlusions mentioned in lines 124-125 do result in potentially strong non-linearities (that are btw also difficult to model with standard manifolds). A new object may suddenly appear. How would your approach react? Connected to this and sparse coding, the linearity assumption of standard sparse coding may be regarded as one reason for the sparse coefficients being non-smooth. After all, sparse coding adds and substracts basis functions with the objective of reconstruction. If occlusions play a role, alternatives to linear approaches (e.g. sigmoid belief nets or explicit occlusion models (Jojic and Frey CVPR 2001; Williams and Titsias Neural Comp 2004; Bornschein et al. PLOS 2013) may make sparse coding activities much smoother and easier to track using pooling. Regarding pooling, I think it is nice that pooling is learned from data (which is a difference to some but not all contributions mentioned below). Regarding the numerical experiments, as stated before, a difficulty is the comparison. Many approaches have addressed the problem of learning more structured representation not least deep neural network approaches. Considering the results of Fig. 5, I was missing discussions with some other relevant contributions. Also, the fields that look very localized may actually be caused by the here implicitly assumed binary units, but I am not sure (compare fields in Bornschein et al., 2013). Curved fields are nice but they are also not necessarily a result of hierarchical coding (Olshausen SPIE 2013, Sheikh, Lucke, NIPS 2016 and refs therein). What I was maybe missing most, though, were discussions of hierarchical coding and pooling, e.g., as suggested by Hinton et al., ICANN 2011, or Sabour et al., NIPS 2017. Also style and content models (e.g., Tenenbaum & Freeman, Neural Comp. 2000, Berkes et al., PLOS 2009) discuss results closely related to Fig. 3. In general, could you think of any quantitative evaluation of your approach, e.g., compared to these contributions? In summary, I think addressing of a very challenging task has to be appreciated, and the simultaneous learning of manifold learning and sparsity with feature slowness on video is challenging. The evaluation of the approach and comparison with others is very difficult. UPDATE (after author responses): Answers to my questions were given partly. I had hoped for a suggestion of a more concrete metric for comparison with competing work, though. My concern, therefore, remains what to make of the approach especially in comparison with others. I agree with two other reviewers that the paper is difficult to follow but it also addresses a topic which is difficult to present. Still, the authors should be very motivated to further clarify their approach. I am not more confident or more convinced after the response. Promised additional discussion of previously work will be important, clarification / discussion of binary latents vs. continuous latents, and the other discussed points etc.

Reviewer 3



Summary ------- This paper presents a framework which makes use of manifold learning to add structure into sparse coding models. By adding a manifold structure in the space of the space of the dictionary atoms, it permits to represent each data points with sparser coding signals as the dictionary can be adapted to the specific data point inside the manifold. This idea is combined in section 3 with a temporal regularization for the learned representation which ensure that the sparse code in a video are not changing too fast. Overall assessment ----------------- Overall, while this paper takes a pedagogical approach to the general concepts of sparse coding and manifold learning, it lacks of proper formalism. The objects in equation (1) to (5) are not defined at all. The spaces in which x, phi, alpha and other variables are defined are not given, making it very hard to follow in the following. Also, the notion of manifold embedding, central in this work is never properly explained. This let the reader guess what are each variable. Moreover, the evaluation of this technique is weak, as only qualitative results are shown on limited dataset (20x20 video with a 100 frames). Also, the reconstructed video displayed in the one used to learn the model. More extensive evaluation of this technique is needed, on larger video and with more quantitative results. Questions --------- - Eq(2-3): If I get it correctly, phi_data =x? - Could you clarify the difference between the phi_LM and the P_LM? - Figure1: If I understand correctly, k should be f in the legend to match the text? -l140: I don't see where the sparsity intervene in section2.1. Could you clarify? -Figure3: The obtain representation beta in figure3B is dense. Isn't your goal to derive a "sparse" transform? Typos and nitpicks ------------------ -l40/43: need citations -l110: a m-dimensional manifold in which space? -l111: a generalized eigenvalue decomposition of what? -l227: 'affinity groupe' Post-rebuttal comments -------------- I have carefully read your rebuttal and the other reviewer comments. Overall, I agree that the idea is interesting and novel but this paper intention is to bring together manifold learning and sparse coding and I think it lacks pedagogy to do so. It is very vague, as also stated by R1 and as the different spaces are not clearly stated, it is hard to perceive which object is a vector or a function and in which dimension. Personally, I find it really confusing and hard to follow. The rebuttal does not seem to acknowledge this as stated with "Sparse coding is pretty standard stuff" and "the notion of manifold embedding is pretty clear in the manifold learning literature" and does not clarify the concepts questioned by R1. I am also confused at the fact that the transform is looking for h-sparse function, which are not expected to be sparse according to the rebuttal. I still recommend rejection, but mainly based on the narrative aspect of the paper. It might be biased by my lack of knowledge in manifold learning but I think the pedagogical part is very important for such a novel approach.

Reviewer 4



Paper 6729: The Sparse Manifold Transform OVERVIEW: The authors present an unsupervised learning algorithm that combines (1) sparse representation, with (2) steerable nature through linear combinations of landmarks, and (3) includes temporal smoothness constraints in the basis of the representation. The authors show that the transform is (a) quasi-invertible for sparse signals, (b) it finds sensible groups in the dictionary, (c) it has interesting nonlinear interpolation properties, and (d) stacking several of these layers complicates the identified features as one goes deeper into the structure. Overall, I find the work very interesting because the proposed method combines sensible assumptions and sensible properties emerge from natural datasets. Results are promising and the properties of this transform should be studied in more detail in the future. RECOMMENDATION: If the authors improve some comments on the methods modelling the relations among sparse coefficients, and make the code available for the scientific community (the last is mandatory) I think the work should be accepted in the conference. MAJOR COMMENTS: The interested reader could find technical details and experimental illustrations too schematic. That is why I think the authors have to make the code available. However, presentation and experiments are fair enough given the length restrictions of a conference paper. In follow up works authors should focus on comparisons to alternative representations (which may not be mandatory in this first report). For instance: Does the proposed transform have consequences in neuroscience?. Does it explain better than other unsupervised learning methods the properties of natural neurons (feature tuning) and their interactions? On the engineering side, how can this representation improve pattern recognition tasks? (as opposed to others). Bibliographic note. In sections 1.1 and 4 the authors mention some literature that identified remaining relations between the coefficients of sparse representations (or linear ICA representations) and criticise that these relations are not learnt but typically imposed (Hyvarinen et al. 01, Osindero et al. 06). The authors also cite the literature that captures these relations through statistical criteria but citation is not consistent in both places L52 and L264 (please cite all references in both places). However, they miss some references about the relation between wavelet-like sensors in V1. In this literature the relation between coeffcients sometimes is learnt statistically as in some of the references already included and also in (Schwartz&Simoncelli Nat. Neurosci. 01, Laparra&Malo Front.Human Neurosci. 15), but other times they emerge from psychophysics and have interesting connections to the PDF of images as in Malo&Laparra Neural Comp. 10. Specifically, Schwartz&Simoncelli01 point out that the referred relations between wavelet-like filters do exist and show that divisive normalization with local interaction between neighbors of the representation may reduce this statistical relations. They learn the neighborhoods and interactions to reduce redundancy. Information maximization after linear transforms also gives rise to the same kind of adaptive nonlinearities where data leads to stronger interaction between coefficients which are closer not only in scale and orientation as reported in [28], but also in temporal frequency or speed (Laparra & Malo 15). Actually, the prinicpal curves method in [28] and (Laparra & Malo 15) heavily relies on the manifold flattening concept referred in [10] and [17]. Interestingly, the psychophysical tuning of these interaction neighborhoods between wavelet-like neurons using divisive normalization also gives rise to bigger redundancy reduction (Malo&Laparra10). MINOR COMMENTS - L227 grouop -> group - I would appreciate more detailed information on how Fig.4 is computed (even though I understand space restrictions). REFERENCES: Schwartz & Simoncelli 2001. Nature Neuroscience 4, 819–825. doi: 10.1038/90526 Laparra & Malo 2015. Frontiers in Human Neurosci. https://doi.org/10.3389/fnhum.2015.00557 Malo & Laparra 2010. Neural Computation 22(12): 3179-3206